# Profiling of Essential Oils from the Leaves of *Pistacia lentiscus* Collected in the Algerian Region of Tizi-Ouzou: Evidence of Chemical Variations Associated with Climatic Contrasts between Littoral and Mountain Samples

**DOI:** 10.3390/molecules27134148

**Published:** 2022-06-28

**Authors:** Chabha Sehaki, Nathalie Jullian, Elodie Choque, Rebecca Dauwe, Jean Xavier Fontaine, Roland Molinie, Fadila Ayati, Farida Fernane, Eric Gontier

**Affiliations:** 1BIOPI-UPJV Laboratory UMRT BioEcoAgro INRAE1158, SFR Condorcet FR CNRS 3417, UFR de Sciences, Université de Picardie Jules Verne, 33 Rue Saint Leu, 80000 Amiens, France; chabha.sehaki@etud.u-picardie.fr (C.S.); nathalie.jullian@u-picardie.fr (N.J.); elodie.choque@u-picardie.fr (E.C.); rebecca.dauwe@u-picardie.fr (R.D.); jean-xavier.fontaine@u-picardie.fr (J.X.F.); roland.molinie@u-picardie.fr (R.M.); 2Laboratory of Natural Resources, University Mouloud Mammeri of Tizi-Ouzou, Tizi Ouzou 15000, Algeria; fadila.ayati@ummto.dz (F.A.); farida.fernane@ummto.dz (F.F.)

**Keywords:** *Pistacia lentiscus*, essential oil, antioxidant activity, fatty acid

## Abstract

Leaves of *Pistacia lentiscus* were collected from two Algerian sites in the mountains and the littoral of the Tizi-Ouzou region. The harvest was conducted in four consecutive seasons on the same selected set of trees. Essential oils (EOs) were extracted by hydrodistillation; then, they were analyzed by gas chromatography coupled mass spectrometry (GC-MS). Forty-seven constituents could be detected and quantified, including α-pinene (2–13%), β-caryophyllene (8–25%), β-myrcene (0.3–19%), bornyl acetate (0.8–7%), δ-cadinene (3–8%), bisabolol (1–9%), β-pinene (0.9–7%), caryophyllene oxide (4–9%), and α-cadinol (3–11%). Antioxidant (AOx) activities of the EOs were assessed by ferric reducing antioxidant power (FRAP), 2,2-diphenyl-1-picrylhydrazyl (DPPH), and 2,2′-azino-bis (3-ethylbenzothiazoline)-6-sulfonic acid (ABTS) assays. Significant differences in EO composition and AOx activities appeared dependent on the season and the site. Variations of AOx activities were significant for the FRAP and ABTS tests but not for DPPH. Characterization of the leaf fatty acyl (FA) profiles was performed by GC-MS. Variability appeared according to season and altitude. Polyunsaturated fatty acids levels were high (27–55%) at the coldest date and place. The levels of linolenic acyl in the leaves were significantly correlated with bisabolol levels in the EOs (Spearman’s correlation coefficient: 0.818). Such results will be useful for the sustainable local valorization of wild *P. lentiscus*. These data also open new routes for further studies on terpenoid biosynthesis using correlation networks and fluxomic approaches.

## 1. Introduction

Belonging to the Anacardiaceae family, *Pistacia lentiscus* (*P. lentiscus*) is one of at least 11 plant species of the *Pistacia* genus. Commonly called “lentisk” or “mastic tree” (in reference to its resin or so-called mastic), *P. lentiscus* is an evergreen shrub, largely distributed in Mediterranean countries [1]. It is a branched perennial thermophile shrub, measuring one to five meters in height. Its leaves are pinnate and evergreen. It is a dioecious species with distinct male and female plants harboring unisexual flowers. These flowers are approximately three millimeters wide and appear in spring (from March to May). They are aromatic, either male or female, and occur in small clusters. The female flowers are yellowish green. The male flowers are dark red. On female trees, the fruit is a rounded mono-sperm drupe of four to six millimeters in diameter. The fruit is first green, then red, and becomes blackish when ripe in autumn [2]. A resin, also called mastic, is the best-known product of this plant. It is an aromatic and resinous substance that seeps from the trunk and the main branches [3]. The traditional use of different parts of the plant for therapeutic applications makes lentisk considered as a medicinal plant [4]. People in rural areas use the infusion of its leaves to treat gastric ulcers [5], respiratory disorders [6], and diabetes for its hypoglycemic effect [7]. They also use decoctions of lentisk aerial parts to treat tooth ache and throat infections [8]. The results of an ethnopharmacological study on lentisk, carried out in Algeria, have led to various therapeutic indications of this plant in the country [9]. The main components and pharmacological properties of the species were reviewed by Nahida (i.e., antimicrobial, hepatoprotective, antioxidant, and anticancer activities) [10] and Bozorgi (i.e., antimicrobial activity) [11]. The *Pistacia* species show variations in morphological, anatomical, chemical, and molecular traits across geographic origins and sexes [12].

Essential oils (EOs) have been produced from several parts of the *Pistacia* species [13]. Extracted from leaves, resin, mature or immature fruits, galls, buds, twigs, or flowers, lentisk EOs are generally a limpid liquid with a yellow color and an intense and piercing aromatic odor [13].

Different EOs from *Pistacia lentiscus* leaves have been produced by hydrodistillation [14]. Their chemical composition greatly depends on their origins (i.e., Algeria, Morocco, Tunisia, Italy, Spain, Egypt, Greece, or France). In Algeria, *P. lentiscus* EOs are mainly characterized by terpinen-4-ol (17%) and limonene (34%) [15,16]. Myrcene (39.2% [17]; 25.3% [18]) is the main compound reported in EOs from Morocco. In Tunisian and Italian EOs, α-pinene represented 9–18% of the total components [19,20]. Other studies reported the presence of monoterpene hydrocarbons [21,22,23] or sesquiterpene hydrocarbons [20,24] as the predominant fraction in lentisk EOs. The change in the chemical composition of the oils has been attributed (i) to the wide range of habitats in which lentisk grows [25]; (ii) to environmental factors; (iii) to seasons [26] and phenological stages [27]; (iv) to gender [2,28].

Lentisk EOs and derivatives have been used in perfumery [29] (e.g., Perfume “Accord Boisé” Minalys™ from the Olfactis Cie), in the food industry as flavoring in drinks [30,31] (e.g., liqueur Callicounis™ and Skinos™), and chewing gum [29] (e.g., Mastic Gum Elma™). *P. lentiscus* EOs have interesting biological activities. Indeed, antimicrobial effects against *Helicobacter pylori*, *Staphylococcus aureus*, *Sarcina lutea*, *Escherichia coli*, *Torulopsis glabrata*, and *Listeria monocytogenes* have been reported [21,29]. The use of *P. lentiscus* EOs as a natural fumigant against Lepidoptera, such as *Ectomyelois ceratoniae* and *Ephesia kuehniella*, had significant insecticidal effects [32]. The oil shows different levels of antioxidant properties [33].

These previous studies described various potential applications of *P. lentiscus* essential oils. Nevertheless, each use was based on the fine composition of essential oils and large variability occurred. It is also known that biological activity depends on the chemical composition of the EOs [34,35]. Most of the time, the main components are important, but sometimes synergic effects due to the different, eventually minor, compounds can lead to surprising enhanced (or decreased) biological activities.

Thus, in order to evaluate the potential of the natural *P. lentiscus* resources from the wide Algerian region of Tizi-Ouzou, it is of primary necessity to describe the profile of the essential oils from the leaves of wild plants. It is also needed to determine the variations of such profiles, and, if possible, to determine the main factors affecting such a profile.

This work aimed to explore the composition of wild *P. lentiscus* EOs extracted from leaves. Sample collection was conducted from two contrasting sites in the Tizi-Ouzou region (i.e., mountain and littoral); on the same selected tree from the bush, sampling was conducted four times a year to cover the four seasons (and yearly phenological stages of this perennial plant). We thus studied the variability of the terpenoid content in *P. lentiscus* and investigated the impact of this variability on its antioxidant activities as a function of both the harvesting period and altitude. In addition, the profiling of the fatty acyls from leaf lipids was performed. It allowed for proving that the main variations in the EOs’ composition were associated with the adaptation of the plants during the colder season in the Tizi-Ouzou Mountains.

## 2. Results and Discussion

### 2.1. Qualitative and Semi-Quantitative Composition of the Essential Oils

The different essential oils (EOs) were analyzed by GC-MS. Based on the previous literature and databases, 47 chromatographic peaks could be annotated (Table 1). These compounds corresponded to chromatographic peaks representing from 88% to 97% of the total area of all peaks. These oils were characterized by their richness in sesquiterpene hydrocarbons. This volatile fraction was the most abundant (29–46%) in almost all the different EOs (Table 1). Monoterpene hydrocarbons were present (5–42%) in all the samples. Oxygenated sesquiterpenes and oxygenated monoterpenes were in 13–37% and 3–9%, respectively. Within all samples, the major compounds detected were: α-pinene (2–13%), β-caryophyllene (8–25%), β-myrcene (0.1–19%), bornyl acetate (0.8–7%), δ-cadinene (3–8%), bisabolol (1–9%), β-pinene (0.9–7%), caryophyllene oxide (4–9%), and α-cadinol (3–11%).

Recent data from different Mediterranean countries have noted such a wide spectrum of chemical variability involving both the main compounds and the total amounts of the terpene classes [36,37]. For terpene classes, monoterpene hydrocarbons usually represented the main fraction: 75% in Egypt [37], 68% in Greece [3], 59% in Tunisia [24], and 56% in Italy [36]. However, in Tunisia, lentisk EOs were equally rich in monoterpene hydrocarbons (41%) and sesquiterpene hydrocarbons (40%) [19]. Lo-Presti et al. [20] and Llorens et al. [31] showed the prominence of sesquiterpenes in EOs from Italy (47%) and Spain (39%). Beyond terpene class variations, terpene levels varied as well. Thus, depending on the soil quality, different EO chemotypes have been described. In Spain, Llorens et al. [31], detected the presence of germacrene (10–17%), β-caryophyllene (11–12%), myrcene (1–11%), and terpinen-4-ol (5–6%) from calcicolous and siliceous soils. In Italy, Congiu et al. [38] detected β-caryophyllene (31%), germacrene (12%), and δ-cadinene (6%) as the main terpenes, while α-pinene and terpinen-4-ol were the chemotypes described in the (also Italian) EOs analyzed by Negro et al. [29]. EOs from Morocco varied in chemical composition; some were rich in myrcene (33–39%) [17,18,39], while others were mainly characterized by their 24% α-pinene level [40]. A number of other chemotypes rich in car-3-ene were also reported in Morocco [41]. The lentisk EOs in Tunisia were characterized by their high level of α-pinene (17%), terpinen-4-ol (12%), and δ-terpinene (9%) [24]. In addition, in Tunisia, germacrene D (11%), α-pinene (9%), limonene (8%), β-caryophyllene (8%), and δ-cadinene (8%) were found to be major compounds by Aissi et al. [19]. The (leaf) EOs from Egypt were described as rich in α-phellandrene [37]. It is thus clear that EOs from the leaves of *Pistacia lentiscus* presented variable compositions depending on the countries where the plants grew. Monoterpenes (i.e., α-pinene, terpineol, myrcene, limonene, and terpinen-4-ol) and sesquiterpenes (i.e., β-caryophyllene, δ-cadinene, and germacrene D) were the most common compounds. However, their levels greatly varied depending on the origin of the samples. The main factor contributing to this chemo-variability was attributed to the environmental conditions in the habitats of the *Pistacia lentiscus* species. Few or no data exist concerning genetic traits associated with the EO profiles. If *P. lentiscus* is an allogamous (dioic) species, we can assume that chemotypes were mostly (and naturally) selected essentially based on pedoclimatic traits. In our study, according to the harvest time (season) and place (altitude), various main chemotypes were detected. Except for the precise percentages, some of our main chemotypes correspond to other profiles previously described: caryophyllene [31,38], δ-cadinene [19], myrcene [18,39], and α-pinene [22]. Variations in EO profiles can be of interest. Nevertheless, it is important to determine the origins of variations before expecting a standardized production and a further economical use of such natural products, hence our interest in this littoral and mountain study.

### 2.2. Quantitative Variations of the Major Compounds and Terpenic Classes

The relative abundance of each compound in the different essential oils (EOs) was determined based on chromatographic peak areas measured in the total ion current (TIC). As previously described [2], each relative abundance was then expressed as a percentage of the peak total area.

Table 1 presents the mean contents (and standard deviations) of each EO based on their origin: season and geographical site. Variations among sample origins clearly occurred. For the main chromatographic peaks, major differences can be observed (Table 1). As such, the highest percentages of α-pinene (12.6%) and β-pinene (7.1%) were, respectively, found in spring and summer at the mountain site. The highest amount of β-myrcene (19%) was detected in the mountains in autumn. In all EOs, the levels of β-caryophyllene exceeded 8% at both sites and harvest periods from 8.8% (winter, mountain) to 25.4% (summer, coast). δ-Cadinene (8.5%) and bisabolol (9.7%) harbored their highest levels in the EOs collected in winter in the mountains. In addition to these compounds, other molecules (also formatted in bold in Table 1) presented, at least once, a level superior to 5%, e.g., bornyl acetate (0.8–7.0%), caryophyllene oxide (4.5–9.9%), and α-cadinol (3.7–11.9%).

A two-factor analysis of a variance ANOVA test was performed with all the values to evaluate the effects of site altitude (Factor 1), season (Factor 2), and the possible interaction between both factors and each molecule. The significance of the effects are presented in Table 1 (column: Altitude * Season Interaction) with the following symbols: *** *p*-value < 0.001; ** *p*-value < 0.01; * *p*-value < 0.05. Most of the time, altitude and/or season had a significant (*p* < 0.05) or a highly significant *p* < 0.01 to *p* < 0.001) effect on the variation of the EO component levels. The interaction between altitude and season (Table 1, column: Altitude * Season Interaction) had a highly significant effect (*p* < 0.001) for most of the molecules found in the different essential oils.

A Tukey test (post hoc) was applied for all the data. Significant differences (*p* < 0.05) between the mean levels of each compound are reported in each row of Table 1 as different letters. Homogeneous groups of the EO contents appeared among the eight altitudinal and seasonal conditions. These data confirm that differences and similarities occurred in the chemical profiles of our EOs. However, these data were not clear enough to determine the main rules on the implied variations in the individual compounds. The same analysis performed on the different terpene classes also showed variations over the four distinct classes of compounds (i.e., Table 1: monoterpene hydrocarbons (MHs); sesquiterpene hydrocarbons (SHs); oxygenated monoterpenes (OMs); oxygenated sesquiterpenes (OS)). MHs were higher in the mountains (28–42.8%) than in the littoral (6.4–27.1%), except for during winter (5.8%). For both sites, the highest rate of MHs was observed in autumn. MH levels in the mountains in autumn (42.8%) were one-and-a-half times the level of MHs from littoral samples in autumn (27.1%). For each site, the differences between levels of MHs were significant (*p* < 0.05) according to the season, except between spring and summer for both sites. Contrary to MHs, sesquiterpenes hydrocarbons (SHs) were higher (*p* < 0.05) in the littoral area (37.9–46.5%) than in the mountain (29.1–33.8%). SH levels were higher in the winter (mountain) and in the summer (littoral). Whatever the site or altitude, OS levels were higher in winter (mountain, 37.0%; littoral, 30.3%). OMs were present in smaller quantities than those of MH-SH-OS for all samples. Their variations were not significant according to the season and the altitude. For the four compound classes, the interaction between altitude and season (Table 1, Altitude * Season Interactions) had a significant (*p* < 0.05) or very significant (*p* < 0.001 for MH and SH) influence on the average levels. According to these results, we can conclude that the two parameters studied (i.e., altitude and harvesting season) contributed significantly to the variation in the chemical composition of the essential oils obtained from leaves of *Pistacia lentiscus* collected in the Algerian region of Tizi-Ouzou. Nevertheless, within all the variability detected for the composition of the different EOs, it was necessary (i) to determine the main traits of each oil and (ii) to compare them with other samples previously described in the literature. As such, the major EO constituents that were detected in lentisk leaves harvested at the vegetative overwintering stage (February in Turkey) were previously described as terpinen-4-ol (29%; [42]) and β-myrcene (39%; [17]). For leaves collected at the flowering stage, in different places (i.e., countries), various EO chemotypes have also been described. The major compounds described were: α-pinene (25%), terpinen-4-ol (21%), δ-cadinene (11%), and β-caryophyllene (19%) by Negro et al. [29]; germacrene D (22%), α-pinene (17%), and sabinene (15%) by El Idrissi et al. [43]; tricyclene (7%), terpinen-4-ol (7%), sabinene (6%), and caryophyllene (6%) by Haloui et al. [23]. At a late fruiting stage (i.e., ~October), Mecherara et al. [16] detected the predominance of limonene (43%), α-pinene (34%), and myrcene (33.1 %) in EOs from lentisk leaves collected in Algeria. However, Ismail et al., (2012) in Tunisia reported the presence of α–pinene (20%), limonene (15%), β-pinene (9%), and germacrene D (8%) [22]. At early fruiting (i.e., ~June), the main EO compounds from lentisk leaves growing in the west Algerian region of Oran were terpinen-4-ol (41%), α-terpineol (7%), α–pinene (9%), limonene (9%), β-myrcene (10%), and p-cymene (8%) [44]. When comparing the composition of our EOs with those previously described in the literature and according to the harvest period, some similarities and differences can be raised. We note that some compounds, such as α-pinene, were similar at the flowering and early fruiting stages as was the case of Negro et al. [29], El Idrissi et al. [43], Haloui et al. [23], and Hamiani et al. [44]; similarities also occurred with β–myrcene in the late fruiting stage as described by Mecherara et al. [16], and β-pinene in the early fruiting stage was a main compound as shown by Ismail et al. [22].

However, our results largely differ from those of Zaouali et al. [2], who studied variations in the EOs’ chemical composition according to the plant sex for different harvest periods in Tunisia. In female trees, α-limonene was mostly detected in leaves (26–29%) sampled at the early fruiting (August) and late fruiting (October) stages. Germacrene-D was dominant in leaves (20%) at the flowering and early fruiting stages, and δ-cadinene (15.6%) reached its highest content in leaves in the flowering (March) stage. In male trees, the highest amounts of germacrene-D (13%) were recorded in leaves at the flowering stage. Oppositely, for the variations in our results according to the (limited) literature regarding lentisk EOs, the influence of ecological factors may support an important part of the explanation. In other Mediterranean evergreen plant species, the influence of ecological factors on the chemical composition of EOs extracted from leaves has been clearly pointed out in *Juniperus oxycedrus* [45], *Rosmarinus officinalis* [46], *Myrtus communis* [47], and *Pinus halepensis* [48]. However, for *P. lentiscus*, no detailed information has until now been available to describe, and maybe explain, the precise relationship between environmental factors and the variations in EO composition. To our knowledge, only one unique study described the influence of the altitudinal gradient on the chemo-variation of *P. lentiscus* EO composition [26]. The samples were collected in Algeria, but with only one harvest period (September 2008). Data from our multisite/multiseason results are in accordance with Ait Said et al. [26]. The relationship between altitude and the chemotypes were commonly characterized by caryophyllene and by significant levels of monoterpene and sesquiterpene hydrocarbons.

### 2.3. Multivariate Analysis of All the Components Detected in the EOs

In order to better show the differentiation of the chemical compositions between the different EOs, a principal component analysis (PCA) was carried out on the 47 major compounds annotated above (Figure 1). The two axes of the PCA show 78.6% of the variation. The score plot (Figure 1A) on Axis 1 separates the samples of the two groups according to the sampling site. In each group, the samples for seasons are separated based on Axis 2 of the PCA. Winter and autumn appear mainly separated by Component 2. The mountain group occupies a larger space than the littoral one (i.e., distances between the four seasons were higher for the mountain EOs). This indicates that the variations in the compositions according to the harvesting season were higher at the mountain than at the coastal site. The loading plot (Figure 1B) shows that the most discriminating compounds were the “major” compounds, which appeared at least once at a level greater than 5%. Observations from the littoral samples, regardless of the season, were characterized by high quantities of β-caryophyllene (Group 1). The highest proportion of β-caryophyllene was found in the summer. The winter samples from the littoral group were characterized, in addition to β-caryophyllene, by significant contents α-cadinol and caryophyllene oxide (Group 2). EOs from the mountain site contained high amounts of δ-cadinene, bisabolol, and bornyl acetate in winter (Group 3). Samples collected in autumn had a high proportion of β-myrcene followed by α-pinene (Group 4), mainly from the mountain samples. The samples harvested in spring and summer at the mountain were characterized by high amounts of α-pinene and β-pinene (Group 5).

According to the score and loading plots, the compositions of the EOs analyzed could be subdivided into five sub-chemotypes: Groups 1, 2, 3, 4, and 5.

A difference in altitude is usually accompanied by changes in a range of environmental conditions such as temperature, water precipitation, wind exposure, sunlight intensity, UV radiation, and air humidity [49]. Climatic conditions at (relatively) high altitudes are mainly lower average temperatures, increased thermal amplitude between day and night, and higher light intensity. Such environmental conditions cause plants to change their morphology, physiology, and productivity in order to protect themselves and adapt to such stressful conditions [49,50].

Abreu and Mazzafera [51] reported that lower temperatures at high altitudes can induce an overproduction of phytochemicals (*Hypericum brasiliense* Choisy). Previous studies have suggested that increasing altitude and subsequent changes in solar radiation and temperature in plant habitats can strongly correlate with the content of secondary metabolites, especially phenolics, due to the fact of their protective function against oxidative damage [52]. In order to evaluate a potential link between our EOs’ profiles and their antioxidant properties, further studies were performed with a primary focus on antioxidant activity measurements.

### 2.4. Antioxidant Activity

The antioxidant activities (AOx) of the different essential oils, assessed by the ferric reducing power (FRAP), free radical scavenging activity (DPPH), and the ABTS radical cation reduction tests are presented in Table 2.

According to the origin of the essential oil (i.e., harvesting season and altitude), ANOVA analysis showed that the altitude and season interaction had a significant effect (*p* < 0.001) on variations in antioxidant activity. Significant variations (*p* < 0.05) in the different antioxidant activities can be observed using the post hoc test (Table 2). The FRAP test always led to higher values than the ABTS and the DPPH tests. As compared with the littoral samples, the highest antioxidant (AOx) effect was measured for the mountain samples among the three tests. The ferric reducing capacity (FRAP) of these EOs varied from 9.6 (spring) to 22.3 mgTE/gEO (summer) for the mountain site and from 6.4 (winter) to 15.7 mgTE/gEO (summer) for the littoral samples. Four conditions could be statistically separated (annotation as “a, b, c, and d” for the post hoc test in in Table 2, column: FRAP). Some of the highest FRAP values were recorded for samples collected during the summer period at both sites. However, they were higher at the mountain site (22.3 mgTE/gEO versus 15.7 for littoral). The antioxidant capacity determined by the two other tests, DPPH and ABTS, was quite comparable (but lower than FRAP). The extreme ABTS values were obtained from the mountain site. They were between 0.09 (summer) and 0.32 mgTE/gEO (spring). The post hoc test separated five significantly different homogeneous groups (annotated as “a, b, c, d, and e” in Table 2, column: ABTS). For the DPPH values, levels between 0.06 (winter, littoral) and 0.44 mgTE/gEO (spring, mountain) were found. Nevertheless, in the case of DPPH activity, the post hoc test distinguished significant differences (i.e., “a, b, c, and d”) between all the sample origins (altitude and season).

The antioxidant values of vitamin C (used as a positive control for validating the quantification method) showed that 1 g of Trolox reacted as ~1 g of vitamin C (i.e., 1.08, 0.89, and 1.34 gTE/g vitamin C for the FRAP, ABTS, and DPPH, respectively; Table 2). This confirmed that the measurements of antioxidant activity were correct.

According to the best knowledge of the authors, no report on this plant is available; therefore, comparisons were made with other species from the same genus. Studies on the antioxidant activity of essential oils extracted from the *Pistacia* species are not as numerous. Barra et al. [53] showed a low antiradical capacity of essential oils obtained from *P. lentiscus* leaves collected from different origins and harvesting periods. In addition, Bouyahya et al., showed fairly good antioxidant effects (ABTS and DPPH tests) by the EOs from whole aerial parts of *P. lentiscus* harvested in Morocco [39]. In addition, EOs from *P. atlantica* [13] presented a low capacity for free radical neutralization and an interesting reducing power. On the other hand, Bampouli et al. [54] showed that *P. lentiscus* var. chia EOs did not exhibit antioxidant activity with the DPPH test.

An exploration of the potential correlations between AOx activities and the levels of each component in the EOs was undertaken. Table 3 shows the correlation coefficients between the FRAP, ABTS, and DPPH values and the components previously identified as the main compounds in EOs, as far as the sum of the oxygenated monoterpenes and oxygenated sesquiterpenes. According to this correlation analysis (Table 3), several observations can be noted. The compounds contributing most to the variation in antioxidant activities among the major compounds of EOs were: (1) bisabolol, δ-cadinene, and ß-pinene for the FRAP test; (2) β-caryophyllene, bornyl acetate, caryophyllene oxide, α-cadinol, and oxygenated monoterpenes for the DPPH test; (3) caryophyllene oxide and α-cadinol for the ABTS test.

The change in antioxidant activity can be explained by the variation in the chemical composition of the EOs. However, it is nevertheless difficult to attribute these differences to a single component [53]. In general, strong correlations between individual metabolites and the antioxidant activity of plant extracts are rarely observed, even if multivariate models can show the predictive power of the metabolome for antioxidant activity and can indicate which groups of metabolites contribute most to this activity [55,56]. Sometimes, in addition to the major compounds, minor compounds can also play a significant role in the AOx activity of an EO. In fact, the reducing activity of EOs is obviously related to the oxygenated compounds and the terpenes with conjugated double bonds in these EOs [57].

### 2.5. Profiling of the Fatty Acids from the Leaf Acyl Lipids

Because we suspected a main link between climatic environment (i.e., sites and seasons) and EO composition, we decided to evaluate the fatty acyl (FA) profile of the leaves, with a more dedicated view toward the level of FA unsaturation, which is commonly associated with a physiological adaptation of plant tissues to lower temperatures. This was decided in order to correlate symptoms of low temperature exposure (fatty acid desaturases overexpression) to specific EO profiles, even if we do not suspect a direct metabolic link associating the terpene and fatty acid synthesis (i.e., no presumable fluxomic correlation). It was chosen to extract total lipids from leaves and to trans-esterify all the esterified acyls (from acyl lipids) using a TMAH reactant [58]. The resulting fatty acid methyl esters (FAMEs) were analyzed by GC-MS. The FA composition of the different leaves is presented in Table 4.

Ten FAMEs were identified (Table 4). They represented 94 to 97% (peak area % in TIC) of the extracts. The leaves were all characterized by important levels of polyunsaturated fatty acids (PUFAs): 27–55% linoleic and linolenic FA. The highest PUFA levels were detected in winter in the mountains (55%: linoleic 20% and linolenic 35%). Saturated fatty acids (SFAs), mainly palmitic and stearic FAs, reached 34 to 39%. The monounsaturated oleic acid varied between 3 and 31%. The relative amount of the FA classes were in the order PUFAs > SFAs > MUFAs. The main fatty acids detected in the leaf lipids were linolenic acid (17–35%), palmitic acid (24–31%), linoleic acid (9–28%), and oleic acid (2–30%).

In comparison with the few previous studies, our investigation is the first description of the variation in FA contents in leaves of Algerian lentisk according to the period and altitude at harvest. Our quantitative results are mostly comparable to those of Diamantoglou et al. [59], obtained in Greece, and those of Akdemir et al. [60], in Turkey. Our lentisk leaves from the Algerian Tizi-Ouzou region presented FA proportions comparable to those of the Greece and Turkish ones for C14:0 (myristic), C16:0 (palmitic), C16:1 (palmitoleic), C18:2 (linoleic), and C18:3 (linolenic). On the other hand, behenic (C22:0) was not detected in the samples from Greece and Turkey. Furthermore, the results of Harrat et al. [61] (*P. lentiscus* from Algeria) reported the presence of other minor fatty acids in the leaves such as C12:0, C13:0, C14:1, C15:0, and C20:0. We did not significantly observe the presence of these minor FAs in our samples. In our quantitative approach, the interaction between altitude and harvesting season was tested (Table 4, column: Altitude * Season Interaction). It had a significant effect on the variation of the levels of the main FAs. This was not true for C16:0, C18:0, and C22:0. This interaction was also significant on the variation of saturated fatty acid (SFA) and polyunsaturated fatty acid contents (*p* < 0.01). It was less significant for monounsaturated fatty acids (MUSFAs) (*p* < 0.05). The ANOVA results showed that the seasonal effect had more impact (*p* < 0.001) on the variation in the FA contents than the altitude effect. In order to develop a better view of our results, we performed principal component analysis (Figure 2).

### 2.6. Multivariate Analysis of the Fatty Acids Detected (as Methyl Ester: FAMEs) in the Leaf Acyl-Lipids

The two axes of the PCA analysis of the FAMEs show 94.7% of the variance of the data. The score plot (Figure 2A) separates the samples from mountain and littoral, principally, for the most extreme conditions (i.e., mountain in winter and littoral in summer) in the first dimension of the PCA. The second dimension of the PCA mainly separates the mountain samples in the summer. The loading plot (Figure 2B) shows that unsaturated fatty acids (i.e., C18:1, C18:2, and C18:3) were the main markers of seasonal and altitudinal variations within the analyzed samples. In the case of monounsaturated fatty acids, the oleic acid content increased significantly during the summer months and decreased again strongly in winter at both sites. However, the levels of this acid were higher at the lowest altitude (i.e., littoral). Polyunsaturated fatty acids (PUFAs) decreased during the warm season and increased in winter at the two growing areas. Linolenic acid (a triunsaturated PUFA, also qualified as an omega 6 fatty acid) reached its maximum (35%) of total FAs during the winter months at high altitude. In contrast, linoleic acid was characteristic of the winter samples at low altitude. During the winter months, PUFAs generally increased [62]. This can be attributed to a higher expression of delta12 and delta15 desaturases genes that drive the respective transformation of (i) oleic acid to linoleic acid and (ii) linoleic acid to linolenic acid [63]. This corresponds to an acclimatization of the plant to lower temperatures [59,62]. At the level of cell membranes, it allows a maintenance of phospholipid fluidity and, thus, of membrane activity. Such an increase in the level of polyunsaturated fatty acyls in plant cell membranes was associated with a progressive decrease in the daily temperatures at the end of autumn and the beginning of winter. It also includes the effects of decreasing temperature during the night. This phenomenon was amplified at higher altitudes, where night temperatures are usually colder (and earlier) compared to littoral areas.

The correlation coefficients between the main FAs in leaves and the main terpenoids from the EOs (Table 5) showed that β-caryophyllene and bisabolol were oppositely associated with linolenic (C18:3) and oleic (C18:1) acyls. When colder conditions occurred, C18:3 and bisabolol levels increased. When medium temperatures occurred, C18:1 and β-caryophyllene levels were higher. There was no direct link established between the biosynthetic pathway of fatty acids and terpenoids, but this correlation clearly demonstrated that the variation in temperatures from the mountain versus littoral site explained most of the variations in the EO compositions. Finally, two main EO chemotypes could be distinguished: a “cold winter chemotype”, characterized by relatively high levels of bisabolol (Group 3 in Figure 2); and a “temperate chemotype” characterized by high levels of β-caryophyllene. The chemotype of temperate samples can be subdivided into three sub-chemotypes: Groups 1, 4, and 5 as discussed above in Figure 2.

## 3. Materials and Methods

### 3.1. Sampling and Botanical Identification

Samples from *Pistacia lentiscus* were collected from plants in the Algerian region of Tizi-Ouzou. Locally called “Wilaya of Tizi-Ouzou”, this region covers approximately 3000 km^2^ (100 km EW; 50 km NS) and is limited at the south side by the Djurdjura Mountains and to the north by the Mediterranean Sea. Its relatively contrasted climatic regimes are representative of those of the larger zone of the north of Algeria in between the coast and the mountains. For this study, we selected two geographical sites in the region of Tizi-Ouzou: The first site was located in Ait-Irane, on a mountain in the Tizi-Ouzou region (longitude 36°29′58.3″ N, latitude 4°04′43.4″ E, altitude 876 m). Its Mediterranean climate is characterized by a dry summer and relatively low temperatures during the winter (Appendix A; [64], see Appendix A). The second sampling site was selected in Tigzirt, on the littoral of the Tizi-Ouzou region (longitude 36°53′43.0″ N, latitude 4°11′00.4″ E, altitude 13 m). Its summer and winter temperatures are less extreme (Appendix A; [65]) and more humidity is present all year long (Appendix A; [65]). The contrast between the night/day temperatures is also lower (Appendix A; [65]). At each site, a set of three individual female trees were chosen for all the different samplings. The average height of each tree was between 3.1 and 3.7 m. The leaves were collected at human height (1.6–2 m, i.e., halfway up the crown of each shrub) in order to present a homogeneous set of leaves. Each leaf sample was taken from 4 branches located at four opposite sites all around each tree. Each sample was taken in a staggered manner so as not to resample areas that had already been taken in the previous–following season. Sampling was conducted in the morning at sunrise (6–8 a.m.). Samples were taken at both sites on the same plants over four seasons in 2019 at four different vegetative stages: vegetative overwintering (mid-February), full flowering (mid-April), early fruiting (mid-August), and late fruiting (mid-October).

The plant species, *Pistacia lentiscus*, was identified using the Flora of Quézel and Santa (1962–1963). Furthermore, samples of lentisk were deposited at the ENSA (National High School of Agronomy in Algeria) herbarium. This identification was confirmed by Benhouhou and this led to the delivery of a certificate (ref: 18112020/22:09/ENSA24112020) attesting that this plant was, indeed, *Pistacia lentiscus*.

### 3.2. Extraction of Essential Oils

Leaf samples were cleaned of debris and air dried in the dark for 7 days at a temperature of 25–30 °C. The essential oils (EOs) were extracted by a hydrodistillation process using a Clevenger-type apparatus. Then, the EOs were stored in opaque glass bottles at 4 °C until further use. According to the period and altitude of the harvest, eight groups of essential oils were obtained: winter mountain, spring mountain, summer mountain, autumn mountain, winter littoral, spring littoral, summer littoral, and autumn littoral.

### 3.3. GC-MS Analysis of Essential Oils

The chemical analysis of the EOs was carried out by gas chromatography mass spectrometry (GC-MS) using a Trace GC-Ultra instrument coupled to a DSQ-II (single-quadrupole) mass spectrometer supplied by Thermo Fisher Scientific (Asnières-sur-Seine, France). The chromatographic separation was conducted on a 30 m × 0.25 mm × 0.25 μm TR-5 MS capillary column (5% phenyl and 95% polysiloxane from Thermo-France). One microliter of sample was injected in the splitless mode at 200 °C. The oven temperature was set at 50 °C for three minutes; then, it was programmed to increase from 50 to 270 °C at a rate of 5/min and then to increase to 330 °C at a rate of 10/min. Helium was used as carrier gas (flow rate 1 mL/min). The transfer line was heated to 280 °C. Ionization in the electron impact mode was conducted at 220 °C with an energy of 70 eV. A full-scan detection for masses (*m*/*z*) between 35 and 350 UMA was performed at a rate of 3000 UMA/sec (4.18 scan per second). Each sample of essential oil was diluted 1000 times in hexane containing pentadecane (1/4000 *v*/*v*) as the internal standard.

The identification of oil components was assigned by comparison of their retention times and retention indices (RIs). Experimental RIs were calculated as the relative retention times of compounds compared with those of (C10–C40) n-alkanes. Comparison of experimental RIs (RIxp) was conducted with RIs from the literature [2,16,18,19,23,29,31,37,44,66] and with those of authentic compounds available in the authors’ laboratory. Moreover, identification was confirmed by comparing the mass spectra of each molecule with (i) those available in the NIST 05 library of the GC/MS data system and with(ii) the mass spectra published [67]. The relative quantification of each compound within a sample was based on the peak relative (%) surface measured with the total ion current (TIC) after normalization of the surface area to the internal standard.

### 3.4. Antioxidant Activity

The antioxidant activity of EOs was assessed using three techniques: the DPPH radical-scavenging test, the ferric reducing antioxidant power (FRAP) assay, and the ABTS radical-cation reduction test. All measurements were spectrophotometry determined (see below) and expressed in Trolox equivalent antioxidant capacity (TEAC). Trolox is the hydrophilic equivalent of vitamin E and it was used as the standard for all three tests. All measurements were performed in triplicate and the results are expressed as Trolox equivalent (i.e., mg of Trolox equivalent/g of essential oil; mg TE/g EO) for all tests (i.e., DPPH, FRAP, and ABTS). Vitamin C (ascorbic acid) was tested as the control for the three tests; the results are expressed as mg Trolox equivalent/g of vitamin C.

#### 3.4.1. Radical Scavenging Test (DPPH)

An essential oil diluted in absolute ethanol (50 µL; 2 mg/mL) was added to 950 µL of freshly prepared ethanolic (EtOH 100%) solution of DPPH (100 mM) and diluted to 10 mM on the test day. The mixture was shaken and kept in the dark at room temperature for 20 min. The absorbance was measured at 517 nm. A blank solution was prepared in parallel: 50 µL of absolute ethanol and 950 µL of DPPH 10 mM. The Trolox standard range (i.e., 0, 5, 12.5, 20, and 25 µM) was prepared in absolute ethanol [19,68].

#### 3.4.2. Ferric Reducing Power Test (FRAP)

The FRAP test was prepared by mixing a 10 mM solution of tripyridyltriazine (TPTZ) in hydrochloric acid (HCl 40 mM), a 20 mM ferric chloride (FeCl_3_) solution, and an acetate buffer (300 mM, pH: 3.6), the proportions of 1:1:10, respectively. The reaction was set up by mixing 900 µL of the above FRAP solution with 50 µL of the ethanol (100%)-diluted essential oil sample (2 mg/mL). The reaction mixture was incubated at 37 °C for 30 min. The absorbance was measured at 593 nm. A blank consisted of 50 µL of absolute ethanol and 950 µL of FRAP solution. The Trolox standard range (i.e., 0, 50, 100, 200 m, and 400 µM) was prepared in absolute ethanol [19,69].

#### 3.4.3. Radical Cation Reduction Test (ABTS)

One hundred microliters of essential oil diluted in absolute ethanol (2 mg/mL) was added to 900 µL of the (100%) ethanolic solution of ABTS. ABTS+ was produced by the reaction of 7 mM of ABTS with 2.5 mM of sodium persulfate (K_2_S_2_O_8_) in distilled water for 16 h in the dark. The concentration of ABTS+ was then adjusted by dilution to an absorbance of 0.7 ± 0.02 (at 734 nm). A blank consisted of 100 µL of absolute ethanol and 900 µL of ABTS solution. The Trolox standard range (i.e., 0, 4, 10, 16, and 20 µM) was prepared in absolute ethanol. After incubation in the dark for 6 min, the absorbance of each sample was measured at 734 nm [70].

### 3.5. Preparation of Lipidic Extracts from Lentisk Leaves and GC-MS Analysis

The lipid extracts from leaves were prepared for four harvest periods and from both sites. The extraction of leaf lipids was performed mainly according to Akdemir et al. [60]. Powdered air-dried leaves (300 mg) were extracted in 1.5 mL of methanol–chloroform (1–2 V/V) for 30 min under stirring (tripentadecanoin was also added at 1 mg/mL as an internal standard (Sigma-Aldrich, Saint-Quentin-Fallavier, France) for further GC-MS analysis). After a 5 min centrifugation at 1100× *g*, the supernatant was treated by adding an equivalent volume of hexane. After stirring the mixture, 8 µL of tetramethylammonium hydroxide was added (TMAH; Sigma-Aldrich). The use of tetramethylammonium hydroxide (TMAH) allowed for the preconversion of fatty acids into fatty acid methyl esters (FAMEs) in order to increase their volatility for GC-MS analysis [58]. After resting, the supernatant was recovered and put in vials for GC-MS analysis. For each sample, the separation of the different compounds was performed on a TR-5 MS (5% phenyl and 95% polysiloxane) capillary GC column (30 m × 0.25 mm × 0.25 μm) from Thermo (Asnières-sur-Seine, France). One microliter of each GC-MS sample was injected in the splitless mode at 230 °C. The detector temperature was 250 °C. A gradient of oven temperatures from 230 °C to 290 °C was used (230 to 245 °C at 15 °C·min^−1^; 245 to 290 °C at 90 °C·min^−1^; 290 °C for 2 min). Helium was used as the carrier gas at a flow rate of 1 mL/min. The ionization in the electron impact mode was conducted with an energy of 70 eV and full-scan detection for masses (*m*/*z*) between 35 and 350 atomic mass unit (AMU). The retention indices were calculated according to a series of n-alkanes (C10–C40). The identification was performed by comparison of their mass spectra with those stored in the NIST 05 library (protocol mainly issuing from [71]).

### 3.6. Statistical Analysis

All determinations were performed in triplicate. The results were then expressed as the mean ± standard deviation of three measurements. Statistical analyses were performed in R (R core team, 2021) [72]. The data were analyzed by comparison of the means using a two-factor analysis of variance ANOVA model of Type III, followed by a Tukey test for the determination of significant differences in concentrations of all the detected compounds in the different samples: EOs (with monoterpenes, sesquiterpenes, and others), trans-esterified lipid extracts (fatty acid methyl esters). For each analyzed compound, the assumptions of the ANOVA model (normal distribution and homoscedasticity of the error terms) were checked using the diagnostic plots generated by the plot() function in R, and, if necessary, square root transformation was applied.

The same procedure was used to assess the variation in the antioxidant activities among samples. The correlations between components of the essential oils and antioxidant activities were estimated using Spearman’s correlation coefficient (rs). The same correlation analysis was conducted for major components of the essential oils and acyl composition. Differentiation between populations (based on their EO and acyl compositions) was assessed by principal component analysis (PCA) using R library FactoMiner R (R Core Team 2020) [72,73].

## 4. Conclusions

This paper showed a large variability in the terpenoid (i.e., essential oils) and fatty acid compositions of leaves from *Pistacia lentiscus* collected at two contrasting natural sites in the Algerian region of Tizi-Ouzou. Wild plants naturally growing at high altitudes (mountain site at 876 m) and at low altitudes (littoral site at 13 m) were selected, and some of their leaves were consecutively sampled four times during their different vegetative–reproductive stages over one year. The extraction of essential oils (EOs) from the leaves and the further gas chromatography analysis allowed for the identification and relative quantification of 47 compounds. Based on the main constituents of the EOs, different chemotypes could be described. EOs from the coastal site corresponded to four harvest periods: winter, spring, summer, and autumn. They were characterized by high amounts of β-caryophyllene (C26). EOs from the mountain site were seasonally variable. In winter, EOs contained high amounts of δ-cadinene (C36), bisabolol (C46), and bornyl acetate (C20). Samples collected in spring and summer were characterized by high amounts of α-pinene (C2) and β-pinene (C5). Samples collected in autumn had a high proportion of β-myrcene (C6) followed by α-pinene (C2). The chemical class of monoterpene hydrocarbons was higher in the mountains. In addition, the highest rate of this class was observed in autumn for both sites. Contrary to monoterpene hydrocarbons, sesquiterpene hydrocarbons were present at higher concentrations in the EOs issuing from the coastal area. The antioxidant (AOx) activity of the different EOs varied according to the harvesting period and the altitude. The highest antiradical effects were observed with the FRAP test. The variation in the AOx activities can partially be correlated to the composition of the essential oils. This study thus showed that the change in altitude and harvesting period affected the chemotypic nature of the lentisk EOs as well as their AOx potential, especially at the highest altitudes. The *Pistacia lentiscus* natural population of the Tizi-Ouzou littoral can thus be used for the production of a quite constant chemotyped EO. Oppositely, because of the seasonal variability, EOs from high-altitude populations may be better suited for obtaining dedicated original chemotypes and for the further cultivation and exploitation of their (potentially therapeutic) potential.

The fatty acyl (FA) profile of the leaf acyl lipids was analyzed by GC-MS after total lipid extraction and transesterification into FA methyl esters (FAMEs). The resulting composition was characterized by its richness in polyunsaturated fatty acids (PUFAs) especially in winter in the mountain. PUFA levels were followed by saturated fatty acids (SFAs) and monounsaturated fatty acids (MUFAs). The amounts of FAs decreased in the order PUFA > SFA > MUFA. The major FAs detected in the leaves were linolenic acid, palmitic acid, linoleic acid, and oleic acid. Unsaturated fatty acids (i.e., C18:1, C18:2, and C18:3) levels significantly varied with season and altitude. The colder the climatic conditions the higher the relative ratio of PUFAs. The variability in the EOs’ chemotypes can be associated with this physiological response of plants to temperatures. The leaves harvested at the coldest season were characterized by the highest levels of bisabolol. The EOs from other sites and seasons were characterized by three other chemotypes in which β-caryophyllene was always the main compound. From these results, a planification of leaf harvest can be conducted for producing EOs with specific chemotypes. It is the first step in the further sustainable and rational valorization of natural resources from the region of Tizi-Ouzou. It is also an opportunity to more deeply study the impact of climatic changes on the resulting terpenoid biosynthesis using correlation networks and fluxomic analysis.

## Figures and Tables

**Figure 1 molecules-27-04148-f001:**
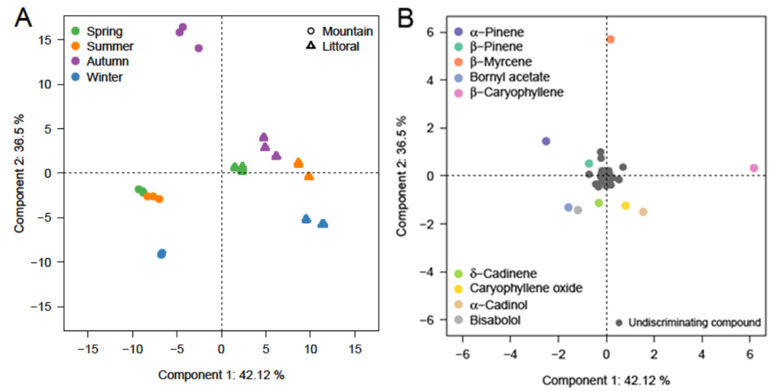
PCA analysis performed on all the compounds from the essential oils of *Pistacia lentiscus* leaf samples collected at the two altitude sites (△ for littoral versus ◯ for mountain) and in 4 consecutive seasons: (**A**) score plot of PC1 versus PC2 scores; (**B**) loading plot of PC1- and PC2-contributing EO compounds (the most discriminating compounds are represented in color).

**Figure 2 molecules-27-04148-f002:**
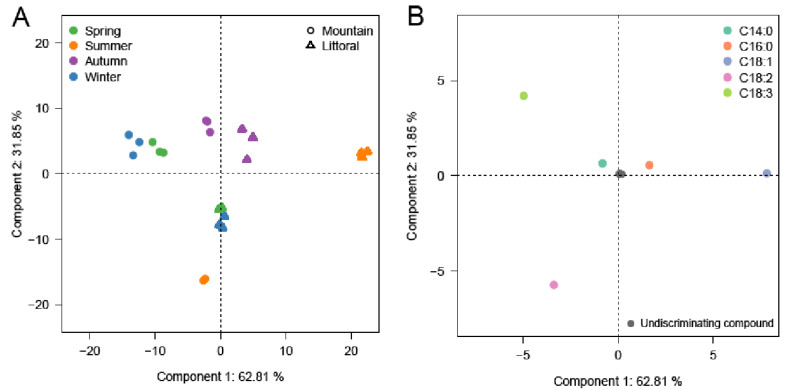
PCA analysis of the FAMEs from the leaf acyl lipids of *Pistacia lentiscus* leaf samples collected at the two altitude sites (△ for littoral versus ◯ for mountain) and 4 consecutive seasons: (**A**) score plot of the PC1 versus PC2 scores; (**B**) loading plot of PC1- and PC2-contributing FAME compounds (the most discriminating compounds are represented in color).

**Table 1 molecules-27-04148-t001:** Compounds detected in the essential oils of *Pistacia lentiscus* leaves collected at two altitude sites and in four consecutive seasons. The levels of each compound are expressed as the chromatographic relative peak surface measured as the total ion current (area % TIC). The extraction yield for each EO was similar: 0.125 ± 0.005 g/100 g of dry matter.

				Means % Content ± Standard Deviation								
				**Mountain**				**Littoral**				**Altitude * Season Interaction**
**C**		**Compounds**	**IR**	**Winter**	**Spring**	**Summer**	**Autumn**	**Winter**	**Spring**	**Summer**	**Autumn**	
**C1**	MH	tricyclene	923	0.2 ± 0.1 *^ab^*	0.9 ± 0.1 *^b^*	0.8 ± 0.1 *^ab^*	0.5 ± 0.1 *^a^*	0.1 ± 0.1 *^c^*	0.3 ± 0.1 *^d^*	0.2 ± 0.1 *^d^*	0.2 ± 0.1 *^ab^*	***
C2	**MH**	**α-pinene**	**935**	**2.6 ± 0.2** ** * ^c^ * **	**13.1 ± 0.1** ** * ^b^ * **	**12.6 ± 0.3** ** * ^b^ * **	**9.4 ± 0.9** ** * ^a^ * **	**2.1 ± 0.4** ** * ^c^ * **	**5.4 ± 0.4** ** * ^d^ * **	**4.9 ± 0.5** ** * ^d^ * **	**8.3 ± 0.3** ** * ^a^ * **	*******
**C3**	MH	camphene	952	0.9 ± 0.1 *^cd^*	3.1 ± 0.1 *^b^*	3.2 ± 0.0 *^d^*	1.6 ± 0.0 *^a^*	0.6 ± 0.1 *^e^*	0.8 ± 0.1 *^f^*	1.1 ± 0.1 *^f^*	1.0 ± 0.0 *^bc^*	***
**C4**	MH	sabinene	975	0.5 ± 0.0 *^e^*	1.4 ± 0.0 *^c^*	1.2 ± 0.1 *^c^*	3.0 ± 0.4 *^a^*	0.3 ± 0.1 *^f^*	1.1 ± 0.1 *^d^*	1.0 ± 0.1 *^cd^*	2.0 ± 0.1 *^b^*	*
C5	**MH**	**β-pinene**	**981**	**0.9 ± 0.0** ** * ^b^ * **	**4.1 ± 0.1** ** * ^a^ * **	**7.1 ± 0.0** ** * ^b^ * **	**3.6 ± 0.4** ** * ^a^ * **	**1.2 ± 0.2** ** * ^b^ * **	**1.2 ± 0.1** ** * ^b^ * **	**3.7 ± 0.3** ** * ^c^ * **	**3.9 ± 0.1** ** * ^a^ * **	*******
C6	**MH**	**β-myrcene**	**993**	**0.1 ± 0.0** ** * ^c^ * **	**0.5 ± 0.0** ** * ^c^ * **	**0.3 ± 0.0** ** * ^c^ * **	**19.0 ± 0.9** ** * ^b^ * **	**0.5 ± 0.1** ** * ^d^ * **	**5.0 ± 0.1** ** * ^b^ * **	**4.8 ± 0.9** ** * ^ab^ * **	**6.1 ± 0.4** ** * ^a^ * **	*******
**C7**	MH	α-phellandrene	1011	0.0 ± 0.0 *^cd^*	0.4 ± 0.0 *^bc^*	0.0 ± 0.0 *^bc^*	0.4 ± 0.0 *^b^*	0.1 ± 0.1 *^e^*	0.1 ± 0.0 *^de^*	0.1 ± 0.0 *^a^*	0.2 ± 0.0 *^a^*	***
**C8**	Oth	o-cymene	1032	0.2 ± 0.1 *^c^*	0.7 ± 0.1 *^ab^*	0.9 ± 0.1 *^bc^*	0.9 ± 0.1 *^ab^*	0.3 ± 0.1 *^d^*	0.3 ± 0.1 *^cd^*	0.5 ± 0.1 *^d^*	0.6 ± 0.1 *^a^*	***
**C9**	Oth	p-cymene	1033	0.1 ± 0.1 *^ac^*	0.2 ± 0.1 *^a^*	0.4 ± 0.1 *^ab^*	0.4 ± 0.1 *^a^*	0.1 ± 0.1 *^bc^*	0.1 ± 0.1 *^a^*	0.2 ± 0.1 *^c^*	0.3 ± 0.1 *^a^*	
**C10**	MH	limonene	1036	0.5 ± 0.1 *^f^*	3.8 ± 0.1 *^f^*	2.7 ± 0.0 *^b^*	4.2 ± 0.1 *^b^*	1.4 ± 0.2 *^e^*	4.7 ± 0.1 *^d^*	1.1 ± 0.2 *^c^*	4.9 ± 0.3 *^a^*	***
**C11**	MH	ɣ-terpinene	1066	0.2 ± 0.0 *^d^*	0.5 ± 0.0 *^c^*	0.3 ± 0.0 *^c^*	0.9 ± 0.0 *^a^*	0.1 ± 0.0 *^e^*	0.3 ± 0.0 *^d^*	0.3 ± 0.0 *^bc^*	0.4 ± 0.0 *^b^*	***
**C12**	MH	α-terpinolene	1094	0.0 ± 0.0 *^c^*	0.2 ± 0.0 *^d^*	0.0 ± 0.0 *^bc^*	0.2 ± 0.0 *^b^*	0.1 ± 0.0 *^d^*	0.2 ± 0.0 *^d^*	0.1 ± 0.0 *^a^*	0.1± 0.0 *^a^*	***
**C13**	Oth	2-nonanaone	1099	0.1 ± 0.0 *^cd^*	0.3 ± 0.0 *^b^*	0.1 ± 0.0 *^bc^*	0.6 ± 0.1 *^a^*	0.1 ± 0.0 *^d^*	0.3 ± 0.0 *^b^*	0.4 ± 0.1 *^a^*	0.5 ± 0.1 *^a^*	***
**C14**	Oth	nonanol	1109	0.2 ± 0.1 *^ac^*	0.4 ± 0.1 *^d^*	0.3 ± 0.1 *^bcd^*	0.1 ± 0.1 *^a^*	0.1 ± 0.1 *^ab^*	0.5 ± 0.1 *^cd^*	0.4 ± 0.1 *^ad^*	0.3 ± 0.1 *^ac^*	
**C15**	Oth	nonanal	1114	0.0 ± 0.0 *^a^*	0.2 ± 0.0 *^b^*	0.0 ± 0.0 *^a^*	0.0 ± 0.0 *^a^*	0.0 ± 0.0 *^a^*	0.2 ± 0.0 *^b^*	0.0 ± 0.0 *^a^*	0.0 ± 0.0 *^a^*	
**C16**	OM	menta-2-en-1-ol <E-p>	1137	0.0 ± 0.0 *^ac^*	0.1 ± 0.0 *^ac^*	0.0 ± 0.0 *^bc^*	0.0 ± 0.0 *^ab^*	0.0 ± 0.0 *^c^*	0.0 ± 0.0 *^c^*	0.0 ± 0.0 *^c^*	0.0 ± 0.0 *^a^*	
**C17**	OM	borneol	1187	0.1 ± 0.0 *^ab^*	0.2 ± 0.0 *^ab^*	0.1 ± 0.0 *^ab^*	0.1 ± 0.0 *^a^*	0.0 ± 0.0 *^ab^*	0.1 ± 0.0 *^c^*	0.1 ± 0.0 *^ab^*	0.1 ± 0.0 *^b^*	***
**C18**	OM	terpinen-4-ol	1207	1.3 ± 0.1 *^ce^*	1.4 ± 0.0 *^b^*	1.4 ± 0.2 *^de^*	1.7 ± 0.1 *^a^*	0.5 ± 0.0 *^e^*	0.9 ± 0.0 *^cd^*	1.5 ± 0.2 *^cd^*	1.5 ± 0.0 *^c^*	***
**C19**	OM	α-terpineol	1209	0.4 ± 0.1 *^d^*	0.4 ± 0.1 *^cd^*	0.9 ± 0.1 *^c^*	0.5 ± 0.1 *^a^*	0.3 ± 0.1 *^c^*	0.5 ± 0.1 *^b^*	0.5 ± 0.1 *^e^*	0.6 ± 0.1 *^ab^*	***
C20	**OM**	**bornyl acetate**	**1297**	**6.2 ± 0.2** ** * ^a^ * **	**7.0 ± 0.1** ** * ^b^ * **	**4.9 ± 0.3** ** * ^b^ * **	**1.5 ± 0.1** ** * ^c^ * **	**2.8 ± 0.1** ** * ^b^ * **	**1.9 ± 0.1** ** * ^f^ * **	**1.9 ± 0.0** ** * ^d^ * **	**0.8 ± 0.0** ** * ^e^ * **	*******
**C21**	Oth	2-undecanone	1301	1.6 ± 0.0 *^d^*	2.0 ± 0.0 *^bc^*	0.9 ± 0.1 *^b^**^c^*	2.1 ± 0.1 *^b^*	1.6 ± 0.3 *^cd^*	2.0 ± 0.1 *^c^*	1.9 ± 0.2 *^a^*	2.5 ± 0.2 *^b^*	***
**C22**	OM	α-terpinyl acetate	1358	0.2 ± 0.0 *^d^*	0.3 ± 0.0 *^f^*	0.0 ± 0.0 *^c^*	0.0 ± 0.0 *^e^*	1.0 ± 0.1 *^a^*	1.5 ± 0.1 *^c^*	0.3 ± 0.0 *^a^*	0.5 ± 0.1 *^b^*	***
**C23**	SH	α-copaene	1386	0.9 ± 0.0 *^b^*	0.8 ± 0.0 *^c^*	1.2 ± 0.0 *^c^*	0.7 ± 0.0 *^a^*	0.3 ± 0.0 *^d^*	0.6 ± 0.0 *^e^*	0.5 ± 0.0 *^g^*	0.4 ± 0.1 *^f^*	***
**C24**	SH	β-bourbonene	1395	0.0 ± 0.0 *^ab^*	0.1 ± 0.0 *^de^*	0.0 ± 0.0 *^ac^*	0.0 ± 0.0 *^e^*	0.1 ± 0.0 *^bcd^*	0.1 ± 0.0 *^ce^*	0.0 ± 0.0 *^a^*	0.0 ± 0.0 *^a^*	***
**C25**	SH	β-elemene	1398	1.1 ± 0.0 *^d^*	1.2 ± 0.0 *^cd^*	1.5 ± 0.0 *^f^*	1.0 ± 0.0 *^e^*	1.4 ± 0.1 *^a^*	1.2 ± 0.0 *^c^*	1.5 ± 0.0 *^ef^*	1.3 ± 0.0 *^b^*	***
C26	**SH**	**β-caryophyllene**	**1435**	**8.8 ± 0.1** ** * ^e^ * **	**10.7 ± 0.2** ** * ^d^ * **	**10.8 ± 0.5** ** * ^f^ * **	**13.2 ± 0.9** ** * ^f^ * **	**24.9 ± 1.0** ** * ^c^ * **	**17.2 ± 0.6** ** * ^b^ * **	**25.4 ± 0.8** ** * ^b^ * **	**21.7 ± 0.1** ** * ^a^ * **	*******
**C27**	SH	β-cubebene	1444	0.8 ± 0.1 *^b^*	0.8 ± 0.0 *^ab^*	0.4 ± 0.1 *^ab^*	0.8 ± 0.0 *^ab^*	0.9 ± 0.1 *^ab^*	0.9 ± 0.1 *^ab^*	0.8 ± 0.1 *^a^*	1.6 ± 0.9 *^ab^*	
**C28**	Oth	isoamyl benzoate	1455	0.8 ± 0.1 *^ab^*	0.7 ± 0.1 *^ab^*	1.0 ± 0.1 *^a^*	0.6 ± 0.1 *^b^*	0.8 ± 0.1 *^ab^*	0.6 ± 0.1 *^ab^*	0.5 ± 0.1 *^c^*	0.6 ± 0.1 *^b^*	***
**C29**	SH	α-caryophyllene	1472	2.1± 0.0 *^f^*	2.0 ± 0.0 *^e^*	2.3 ± 0.1 *^h^*	1.7 ± 0.1 *^g^*	3.3 ± 0.0 *^a^*	2.5 ± 0.1 *^b^*	3.5 ± 0.0 *^d^*	2.8 ± 0.0 *^c^*	***
**C30**	SH	allo-aromadendrene	1477	0.8 ± 0.0 *^a^*	0.7 ± 0.0 *^b^*	0.7± 0.0 *^b^*	0.5 ± 0.0 *^b^*	0.6 ± 0.0 *^a^*	0.7 ± 0.0 *^c^*	0.6 ± 0.0 *^c^*	0.4 ± 0.0 *^d^*	***
**C31**	SH	ɣ-muurolene	1489	2.6 ± 0.0 *^a^*	1.9 ± 0.0 *^d^*	2.8 ± 0.1 *^c^*	1.2 ± 0.1 *^c^*	1.4 ± 0.1 *^b^*	1.7 ± 0.1 *^e^*	1.4 ± 0.0 *^g^*	1.0 ± 0.0 *^f^*	***
**C32**	SH	germacrene D	1498	2.5 ± 0.6 *^bc^*	2.8 ± 0.1 *^d^*	1.2 ± 0.3 *^d^*	4.0 ± 0.1 *^d^*	4.2 ± 0.3 *^cd^*	4.2 ± 0.3 *^b^*	4.2 ± 0.5 *^a^*	3.2 ± 0.2 *^b^*	***
**C33**	SH	α-muurolene	1514	1.9 ± 0.2 *^b^*	1.7 ± 0.0 *^bc^*	2.5 ± 0.0 *^cd^*	1.2 ± 0.0 *^d^*	1.8 ± 0.1 *^a^*	1.6 ± 0.0 *^cd^*	1.7 ± 0.1 *^e^*	1.4 ± 0.0 *^d^*	***
**C34**	SH	β-bisabolene	1520	1.1 ± 0.0 *^a^*	0.8 ± 0.0 *^a^*	0.4 ± 0.0 *^a^*	0.8 ± 0.4 *^a^*	0.4 ± 0.4 *^a^*	0.8 ± 0.2 *^a^*	0.6 ± 0.0 *^a^*	0.6 ± 0.0 *^a^*	*
**C35**	SH	ɣ-cadinene	1530	0.6 ± 0.1 *^b^*	0.8 ± 0.1 *^b^*	0.9 ± 0.0 *^b^*	0.4 ± 0.0 *^c^*	1.1 ± 0.0 *^a^*	0.8 ± 0.0 *^b^*	0.7 ± 0.1 *^c^*	0.6 ± 0.0 *^b^*	***
C36	**SH**	**δ-cadinene**	**1534**	**8.5 ± 0.0** ** * ^a^ * **	**4.7 ± 0.1** ** * ^b^ * **	**3.6 ± 0.4** ** * ^b^ * **	**3.0 ± 0.3** ** * ^b^ * **	**4.8 ± 0.2** ** * ^a^ * **	**4.9 ± 0.1** ** * ^b^ * **	**4.6 ± 0.3** ** * ^a^ * **	**3.3 ± 0.1** ** * ^c^ * **	*******
**C37**	SH	calamenene	1540	2.3 ± 0.0 *^a^*	1.6 ± 0.2 *^c^*	1.5 ± 0.0 *^b^*	0.7 ± 0.0 *^b^*	0.9 ± 0.0 *^a^*	1.0 ± 0.0 *^d^*	0.9 ± 0.0 *^d^*	0.6 ± 0.0 *^e^*	***
**C38**	OS	elemol	1565	0.8 ± 0.0 *^c^*	0.2 ± 0.0 *^f^*	0.5 ± 0.0 *^a^*	0.5 ± 0.0 *^d^*	0.4 ± 0.0 *^e^*	0.6 ± 0.0 *^b^*	0.2 ± 0.0 *^e^*	0.3 ± 0.0 *^g^*	***
C39	**OS**	**caryophyllene oxide**	**1573**	**8.7 ± 0.2** ** * ^c^ * **	**5.7 ± 1.2** ** * ^c^ * **	**7.9 ± 0.7** ** * ^bc^ * **	**4.5 ± 0.4** ** * ^d^ * **	**9.9 ± 0.5** ** * ^a^ * **	**7.8 ± 0.1** ** * ^ab^ * **	**7.5 ± 0.6** ** * ^c^ * **	**7.8 ± 1.0** ** * ^cd^ * **	******
**C40**	OS	humulene epoxide II	1633	0.9 ± 0.1 *^bc^*	0.6 ± 0.1 *^df^*	0.7 ± 0.1 *^b^*	0.2 ± 0.1 *^ef^*	0.9 ± 0.1 *^a^*	0.8 ± 0.1 *^bd^*	0.5 ± 0.1 *^cde^*	0.5 ± 0.1 *^f^*	***
**C41**	OS	cubenol	1648	3.0 ± 0.1 *^a^*	1.2 ± 0.0 *^d^*	2.0 ± 0.1 *^cd^*	0.7 ± 0.1 *^b^*	1.0 ± 0.1 *^a^*	1.4 ± 0.0 *^bc^*	1.2 ± 0.1 *^e^*	0.8 ± 0.0 *^f^*	***
**C42**	OS	epi-Cadinol	1663	2.7 ± 0.0 *^c^*	1.8 ± 0.0 *^cd^*	2.8 ± 0.1 *^de^*	1.3 ± 0.1 *^e^*	2.8 ± 0.1 *^a^*	2.5 ± 0.1 *^b^*	2.7 ± 0.1 *^e^*	2.3 ± 0.0 *^de^*	***
**C43**	OS	δ-cadinol	1668	2.3 ± 0.1 *^b^*	1.1 ± 0.1 *^cd^*	1.3 ± 0.2 *^cd^*	0.6 ± 0.0 *^d^*	1.5 ± 0.1 *^a^*	1.4 ± 0.1 *^bc^*	1.2 ± 0.1 *^cd^*	0.9 ± 0.2 *^e^*	***
C44	**OS**	**α-cadinol**	**1679**	**8.0 ± 0.1** ** * ^cd^ * **	**5.0 ± 0.9** ** * ^bc^ * **	**8.0 ± 0.2** ** * ^bc^ * **	**3.7 ± 0.3** ** * ^d^ * **	**11.9 ± 0.4** ** * ^a^ * **	**7.9 ± 0.1** ** * ^ab^ * **	**7.6 ± 0.8** ** * ^bc^ * **	**9.5 ± 3.2** ** * ^bc^ * **	******
**C45**	OS	aromadendrene oxide	1684	1.0 ± 0.1 *^a^*	0.6± 0.0 *^a^*	1.1 ± 0.4 *^a^*	0.3 ± 0.0 *^a^*	0.5 ± 0.0 *^a^*	0.5 ± 0.0 *^a^*	0.4 ± 0.0 *^b^*	0.3 ± 0.0 *^b^*	**
C46	**OS**	**bisabolol**	**1704**	**9.7 ± 0.3** ** * ^ab^ * **	**2.6 ± 0.0** ** * ^b^ * **	**1.2 ± 0.0** ** * ^a^ * **	**1.4 ± 0.1** ** * ^ab^ * **	**1.3 ± 0.1** ** * ^ab^ * **	**1.5 ± 0.0** ** * ^c^ * **	**1.1 ± 0.1** ** * ^ab^ * **	**1.2 ± 0.0** ** * ^d^ * **	*******
**C47**	Oth	benzyl benzoate	1797	0.4 ± 0.0 *^ab^*	0.2 ± 0.0 *^c^*	0.5 ± 0.0 *^b^*	0.3 ± 0.1 *^c^*	0.8 ± 0.2 *^ab^*	0.9 ± 0.0 *^a^*	0.4 ± 0.0 *^b^*	0.4 ± 0.0 *^ab^*	***
		**Total (sum of C_1-47_)**		88.0 ± 0.6	91.1 ± 1.4	94.8 ± 0.6	94.0 ± 0.3	90.9 ± 0.8	90.9 ± 0.4	94.9 ± 0.3	96.6 ± 3.0	
**Terpene Classes**												
	**MH**	**monoterpene hydrocarbons**		5.8 ± 0.3 *^c^*	28.0 ± 0.1 *^b^*	28.2 ± 0.3 *^b^*	42.8 ± 2.6 *^a^*	6.4 ± 1.1 *^d^*	19.1 ± 1.0 *^c^*	17.4 ± 1.5 *^c^*	27.1 ± 1.2 *^a^*	***
	**SH**	**sesquiterpene hydrocarbons**		33.8 ± 0.5 *^c^*	30.5 ± 0.6 *^c^*	29.8 ± 1.3 *^d^*	29.1 ± 1.4 *^d^*	45.9 ± 0.9 *^a^*	37.9 ± 0.9 *^a^*	46.5 ± 1.7 *^a^*	38.3 ± 0.4 *^b^*	***
	**OM**	**oxygenated monoterpenes**		8 ± 0.3 *^a^*	9.3 ± 0.1 *^a^*	7.3 ± 0.5 *^a^*	3.8 ± 0.2 *^a^*	4.6 ± 0.4 *^a^*	4.9 ± 0.2 *^a^*	4.3 ± 0.3 *^a^*	3.4 ± 0.1 *^a^*	
	**OS**	**oxygenated Sesquiterpenes**		37.0 ± 0.5 *^bc^*	18.8 ± 1.1 *^c^*	25.4 ± 0.6 *^bc^*	13.2 ± 1.0 *^d^*	30.3 ± 0.9 *^a^*	24.3 ± 0.4 *^b^*	22.4 ± 0.9 *^c^*	23.6 ± 4.4 *^e^*	***
	**Oth**	**Others**		3.3 ± 0.1 *^d^*	4.5 ± 0.2 *^cd^*	4.1 ± 0.2 *^bc^*	5.1 ± 0.0 *^ab^*	3.7 ± 0.5 *^d^*	4.8 ± 0.1 *^cd^*	4.3 ± 0.1 *^bc^*	5.1 ± 0.1 *^a^*	

The means followed by different letters (in italic) within the same row were significantly different (*p* < 0.05). The main compounds name and levels are written in bold. Significance effect of the interaction between the altitude and season as determined by ANOVA, *p*-values: *** *p* < 0.001; ** *p* < 0.01; * *p* < 0.05.

**Table 2 molecules-27-04148-t002:** Evaluation of the antioxidant activities of *Pistacia lentiscus* essential oils using FRAP, ABTS, and DPPH tests.

Antioxidant Assays				
Essential Oil Samples		FRAP (mg TE/g EO)	ABTS (mg TE/g EO)	DPPH (mgTE/g EO)
Mountain	Winter	11.3 ± 0.1 *^c^*	0.28 ± 0.01 *^b^*	0.12 ± 0.0 *^a^*
	Spring	9.6 ± 0.3 *^a^*	0.32 ± 0.0 *^c^*	0.44 ± 0.01 *^a^*
	Summer	22.3 ± 0.2 *^c^*	0.09 ± 0.01 *^e^*	0.08 ± 0.01 *^b^*
	Autumn	14.3 ± 0.1 *^a^*	0.24 ± 0.01 *^a^*	0.11 ± 0.0 *^a^*
Littoral	Winter	6.4 ± 0.9 *^c^*	0.09 ± 0 *^c^*	0.06 ± 0.0 *^c^*
	Spring	6.6 ± 1.1 *^b^*	0.23 ± 0.01 *^e^*	0.06 ± 0.0 *^d^*
	Summer	15.7 ± 0.5 *^d^*	0.30 ± 0.01 *^a^*	0.08 ± 0.0 *^b^*
	Autumn	14.8 ± 0.4 *^b^*	0.13 ± 0.01 *^d^*	0.06 ± 0.0 *^c^*
Altitude * Season Interaction		***	***	***
Vitamin C (mg TE/g vit C)		1080 ± 144	894 ± 88	1340 ± 192

Means followed by different letters (in italic) within the same row were significantly different (*p* < 0.05). Significance effect of the interaction between the altitude and season as determined by ANOVA, *p*-values: *** *p* < 0.001.

**Table 3 molecules-27-04148-t003:** Highest correlation coefficients between components of EOs and antioxidant activities.

Major Compounds	FRAP	ABTS	DPPH
α-Pinene	0.37	0.09	0.39
ß-Pinene	0.62 **	−0.09	0.17
β-Myrcene	0.07	0.02	−0.33
Bornyl acetate	−0.18	0.27	0.56 **
β-Caryophyllene	−0.07	−0.18	−0.67 ***
δ-Cadinene	−0.54 **	0.33	0.15
Caryophyllene oxide	−0.24	−0.45 *	−0.46 *
α-Cadinol	−0.11	−0.54 **	−0.56 **
Bisabolol	−0.63 ***	0.36	0.43
Oxygenated monoterpenes	−0.21	0.30	0.55 **
Oxygenated sesquiterpenes	−0.17	−0.40	−0.27

Significant correlations: *** *p* < 0.001; ** *p* < 0.01; * *p* < 0.05.

**Table 4 molecules-27-04148-t004:** FAME analysis of the acyl lipids extracted from lentisk leaves. The levels of each compound are expressed as chromatographic relative peak surface measured as total ion current (area % TIC). The extraction yield for each lipid fraction was similar: 1.95 ± 0.05 g/100 g of dry matter.

		Mountain (Site M)				Littoral (Site L)				Altitude * Season Interaction
**Fatty Acid**	**Corresponding** **acyl**	**Winter**	**Spring**	**Summer**	**Autumn**	**Winter**	**Spring**	**Summer**	**Autumn**	
**Methyl Ester (ME)**										
Myristic acid ME	C14:0	6.6 ± 1.7 *^ac^*	4.0 ± 0.2 *^bc^*	1.8 ± 0.1 *^ac^*	3.4 ± 0.1 *^ab^*	2.5 ± 0.1 *^bc^*	3.2 ± 0.1 *^c^*	2.7 ± 0.3 *^a^*	2.7 ± 0.5 *^d^*	***
Hexadecenoic acid ME	C16:1	1.2 ± 0.3 *^c^*	0.9 ± 0.1 *^b^*	1.2 ± 0.1 *^bc^*	1.1 ± 0.1 *^a^*	0.5 ± 0.1 *^b^*	1.1 ± 0.1 *^ab^*	1.3 ± 0.1 *^bc^*	1.5 ± 0.2 *^bc^*	***
Palmitic acid ME	C16:0	24.8 ± 2.3 *^c^*	28.3 ± 0.3 *^c^*	27.4 ± 0.5 *^c^*	30.3 ± 0.9 *^a^*	26.3 ± 1.1 *^bc^*	31.5 ± 0.4 *^ac^*	31.4 ± 1.0 *^ab^*	31.7 ± 2.0 *^a^*	
Margaric acid ME	C17:0	0.7 ± 0.1 *^c^*	0.8 ± 0.1 *^b^*	0.8 ± 0.1 *^bc^*	0.9 ± 0.1 *^a^*	0.6 ± 0.1 *^b^*	0.9 ± 0.1 *^ab^*	0.9 ± 0.1 *^ab^*	1.1 ± 0.1 *^ab^*	**
Linoleic acid ME	C18:2	20.1 ± 1.7 *^ab^*	20.0 ± 0.2 *^c^*	28.2 ± 8.7 *^a^*	13.9 ± 0.6 *^c^*	26.2 ± 0.9 *^ab^*	22.7 ± 0.5 *^bc^*	9.7 ± 0.2 *^c^*	14.1 ± 1.2 *^bc^*	***
Oleic acid ME	C18:1	2.8 ± 0.2 *^c^*	5.7 ± 0.4 *^bc^*	15.3 ± 8.3 *^d^*	10.8 ± 0.3 *^c^*	14.4 ± 0.3 *^bc^*	10.4 ± 0.6 *^ab^*	30.2 ± 1.2 *^c^*	15.3 ± 1.5 *^a^*	**
Linolenic acid ME	C18:3	35.5 ± 1.8 *^d^*	34.0 ±1.2 *^bc^*	18.5 ± 0.2 *^a^*	31.8 ± 1.1 *^c^*	23.4 ± 0.2 *^e^*	21.7 ± 0.6 *^e^*	17.6 ± 0.7 *^ab^*	27.2 ± 2.8 *^e^*	***
Stearic acid ME	C18:0	1.9 ± 0.2 *^bc^*	2.3 ± 0.1 *^ab^*	2.6 ± 0.1 *^c^*	2.6 ± 0.1 *^a^*	2.1 ± 0.1 *^bc^*	2.2 ± 0.2 *^a^*^b^	2.8 ± 0.1 *^bc^*	2.6 ± 0.3 *^a^*	
Docosanoic acid ME	C22:0	0.9 ± 0.1 *^ac^*	1.2 ± 0.1 *^c^*	1.1 ± 0.1 *^bc^*	1.2 ± 0.1 *^a^*	0.9 ± 0.1 *^bc^*	1.3 ± 0.1 *^c^*	1.2 ± 0.2 *^ac^*	1.2 ± 0.2 *^ab^*	
	Total identified	94.3 ± 3.0	96.7 ± 0.7	96.8 ± 0.7	95.9 ± 0.6	96.9 ± 0.3	94.8 ± 0.9	97.8 ± 0.2	97.3 ± 0.3	
C14; C16; C17; C18; C22	Mean (ΣSFA)	34.8 ± 1.0 *^d^*	36.6 ± 0.4 *^d^*	33.6 ± 0.7 *^d^*	38.3 ± 1.1 *^a^*	32.4 ± 1.2 *^cd^*	39.0 ± 0.7 *^bd^*	39.0 ± 1.5 *^ab^*	39.6 ± 2.9 *^abc^*	**
C16:1; C18:1	Mean (ΣMUFA)	3.9 ± 0.5 *^c^*	6.6 ± 0.4 *^ac^*	16.5 ± 8.3 *^d^*	11.9 ± 0.4 *^bc^*	14.9 ± 0.2 *^ac^*	11.5 ± 0.5 *^ab^*	31.5 ± 1.2 *^c^*	16.8 ± 1.7 *^a^*	*
C18:2; C18:3	Mean (ΣPUFA)	55.5 ± 2.7 *^b^*	54.0 ±1.1 *^bc^*	46.7 ± 8.6 *^a^*	45.7± 0.5 *^bd^*	49.6 ± 0.9 *^bd^*	44.3 ± 1.1 *^cd^*	27.3 ± 0.5 *^bd^*	41.2 ± 1.8 *^d^*	**

Means followed by different letters (in italic) within the same row were significantly different (*p* < 0.05). Significance effect of the interaction between altitude and season as determined by ANOVA, *p*-values: *** *p* < 0.001; ** *p* < 0.01; * *p* < 0.05.

**Table 5 molecules-27-04148-t005:** Correlation coefficients between major components of EOs and FAMEs from the leaves of *Pistacia lentiscus*.

Compounds	C18:3	C18:2	C18:1	C16:0	C14:0
β-Caryophyllene	**−0.58 ***	−0.19	**0.77 *****	0.48 *	0.31
β-Myrcene	0.15	−0.55 ***	0.49 *	0.76 ***	0.08
α-Pinene	0.10	−0.08	−0.18	0.21	0.009
Bornyl acetate	0.29	0.44 *	−0.57 *	−0.68 **	0.3
δ-Cadinene	0.15	0.36	−0.42 *	−0.34	0.39
Bisabolol	**0.81 *****	0.29	**−0.83 *****	−0.37	0.80 ***
ß-Pinene	−0.26 *	−0.10	0.18	0.21	−0.41 *
Caryophyllene oxide	−0.14	0.40	0.09	−0.46 *	−0.32
α-Cadinol	−0.30	0.34	0.21	−0.33	−0.48 *

Significant correlation (in bold): *** *p* < 0.001; ** *p* < 0.01; * *p* < 0.05.

## Data Availability

Data is contained within the article or Appendix A.

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
