# Peer review of "Profiling of Essential Oils from the Leaves of Pistacia lentiscus Collected in the Algerian Region of Tizi-Ouzou: Evidence of Chemical Variations Associated with Climatic Contrasts between Littoral and Mountain Samples"

_molecules, 2022, doi:10.3390/molecules27134148_

Round 1
Reviewer 1 Report
The manuscript presents information about essential oils contents and properties from leaves of Pistacia lentiscus .
The manuscript is well-structured and written. I recommend minor revision of the manuscript before publication.
In my opinion, you should provide further information about Pistacia lentiscus essential oils uses in food industry in Introduction section.
Unnecessary capital letters in the names of chemical compounds.
Provide the same numbers of digits after decimal point in averages and standard deviations in all tables.
Data in Table 1 and Table 4, Obtained results should be presented in absolute data (expressed per g DM), not percentage to compare it.
Detail comment are listed below.
Line 35 Key words, I suggest removing country name and “altitude” from key words.
Line 325 – 327 What do you mean by this sentence “This confirmed that the results of the antioxidant activity of EOs were consistent.”? In my opinion, it confirmed that responses of all methods are similar. Please clarify it.
Line 457 – 471 Could terpenoids be precursors of fatty acids or vice versa? Why did you correlate these values?
Line 502 – 508 3.2. Extraction of essential oil, What was the yield of extracted essential oils? Final content of analyzed compounds strongly depends of this.
Line 569 – 591 What was the yield of lipids extraction?
Author Response
-We have added some more informations (products, trade marks and-or compagnies) for: "In my opinion, you should provide further information about Pistacia lentiscus essential oils uses in food industry in Introduction section."
-We have checked and corrected as demanded in: "Unnecessary capital letters in the names of chemical compounds."
-We have eliminated some digits (when pertinent) as response to: "Provide the same numbers of digits after decimal point in averages and standard deviations in all tables."
-To answer to the remark "Data in Table 1 and Table 4, Obtained results should be presented in absolute data (expressed per g DM), not percentage to compare it.", we have added in Table1 and Table4 these informations respectively:
In Table1...."The extraction yield for each EO was similar: 0.125+/-0.005g/100g of dry matter."
in Table4...." The levels of each compound is expressed as chromatographic relative peak surface measured as total ion current (area % TIC). The extraction yield for each lipid fraction was similar: 1.95+/-0.05g/100g of dry matter."
This facilitates the understanding and the rationale comparison of the values.
-Line 35 In Key words, "country name" and “altitude” have been removed
-for the remark: "Line 325 – 327 What do you mean by this sentence “This confirmed that the results of the antioxidant activity of EOs were consistent.”? In my opinion, it confirmed that responses of all methods are similar. Please clarify it.", we changed the sentences by:
"The antioxidant values of vitamin C (used as positive control for validating the quantification method) showed that 1 g of Trolox reacted as~1 g of vitamin C (i.e. 1.08, 0.89, 1.34 gTE/g vit C for FRAP, ABTS and DPPH respectively; Table 2). This confirmed that the measurements of antioxidant activity were correct."
-We added the precision "The extraction yield for each EO was similar: 0.125+/-0.005g/100g of dry matter." in the text for the question "Line 502 – 508 3.2. Extraction of essential oil, What was the yield of extracted essential oils? Final content of analyzed compounds strongly depends of this."
-We added precision for the question relative with "fatty acids and terpenoids)
we decided to evaluate the fatty acyl (FA) profile of leaves; with a more dedicated view on the level of FA-unsaturation which are commonly associated with a physiological adaptation of plant tissues to lower temperatures. This was decided in order to correlate symptoms of low temperature exposure (fatty acid desaturases overexpression) to specific EO profiles; whether no metabolic link directly associate terpene and fatty acid synthesis (no presumable fluxomic correlation).
-We added the information in Table4...." The levels of each compound is expressed as chromatographic relative peak surface measured as total ion current (area % TIC). The extraction yield for each lipid fraction was similar: 1.95+/-0.05g/100g of dry matter." for the question "Line 569 – 591 What was the yield of lipids extraction?"
Reviewer 2 Report
In this research the authors analyzed essential oils (EOs) from leaves of Pistacia lentiscus. The samples were collected from two sites (mountain and littoral) of the Tizi-Ouzou region in Algeria. The samples harvest and their analysis was done at four consecutive seasons in order to investigate how environmental factors (sites with their climate and seasons) affect the composition of essential oils and antioxidant activity of the main compounds from them. Even though there was a rather wide variability in the results, the authors found correlation between some chemical constituents and sites (mountain and litoral) followed by correlation between essential oil components and their antioxidant activity. Based on that authors proposed to use essential oil from litoral sites (less variability in composition) for the production of quite constant chemotyped EO and essential oil from mountain sites (higher altitude, more seasonal variation more variation in phytochemical composition) for obtaining dedicated original chemotypes (compounds). The study also demonstrated the strong influence of environmental factors on phytochemical composition of essential oils and by that on their bioactivity.
This topic is relevant to the field because it gives rather detailed study how environment impacts the phytochemical composition of essential oils and with their bioactivity. Authors gave new data and detailed study that will enrich the previous published material from this field.
The selected methodology for this research is well chosen and in function to answer the main questions in this study.
The conclusions are appropriate and based on obtained data and address well the main questions.
The references are appropriate and sufficient.
There should be some corrections:
line 57: besides mentioning references for pharmacological properties, authors should mention these properties in brief together with references.
line 109: (Table1.) instead of – table1.) ?
Author Response
-previous line 57 this has been done: "besides mentioning references for pharmacological properties, authors should mention these properties in brief together with references".
-previous line 109: All the "table" have been replaced by "Table"
Reviewer 3 Report
This work explains the qualitative and quantitative variations encountered in the volatile compounds of essential oils of P. lentiscus and their impact on the biological activities of the oils. In my opinion, this work is of great importance.
However some fragments of the Manuscript need to be corrected or clarification.
Overall, the authors use precise and technical English. Some of the language used might be confusing for non-native speakers.
The document contains many missuses of English language, and scientifical writting: missing italics in vitro (see please reference section).
- Compound names should be in tiny (for example “α-pinene” instead “α-Pinene”); check all names.
- check all text: “extracted by hydrodistillation” instead “produced hydrodistillation”
- Line 77: begin the sentence by “indeed, antimicrobial effects against…”
- Line 328: change the sentence by putting that according to the best knowledge of the authors no report on this plant is available; therefore, the comparison will be made with other species of the same genus.
Author Response
-english improvement has been done by MDPI english editing service
-all "in vitro" have been checked and are now in italic (in vitro)
-Compound names are now in tiny (for example “α-pinene” instead “α-Pinene”); all names have been checked.
-for all the text: “extracted by hydrodistillation” now replace “produced by hydrodistillation”
-Line 77: the sentence now begins by “indeed, antimicrobial effects against…”
-(previous) Line 328: "According to the best knowledge of the authors no report on this plant is available; therefore, the comparison will be made with other species of the same genus." now replaces "Comparison of our results to references is not easy because it is one of the 1st times that these results are obtained and can be used as a basis for other experiments. "